# Gone Fishing: Neural Active Learning with Fisher Embeddings

**Jordan T. Ash**
Microsoft Research NYC
ash.jordan@microsoft.com

**Surbhi Goel**
Microsoft Research NYC
goel.surbhi@microsoft.com

**Akshay Krishnamurthy**
Microsoft Research NYC
akshaykr@microsoft.com

**Sham Kakade**
Microsoft Research NYC
University of Washington
sham.kakade@microsoft.com

## Abstract

There is an increasing need for effective active learning algorithms that are compatible with deep neural networks. This paper motivates and revisits a classic, Fisher-based active selection objective, and proposes BAIT, a practical, tractable, and high-performing algorithm that makes it viable for use with neural models. BAIT draws inspiration from the theoretical analysis of maximum likelihood estimators (MLE) for parametric models. It selects batches of samples by optimizing a bound on the MLE error in terms of the Fisher information, which we show can be implemented efficiently at scale by exploiting linear-algebraic structure especially amenable to execution on modern hardware. Our experiments demonstrate that BAIT outperforms the previous state of the art on both classification and regression problems, and is flexible enough to be used with a variety of model architectures.

## 1 Introduction

The active learning paradigm considers a sequential, supervised learning scenario in which unlabeled samples are abundant but label acquisition is costly. At each round of active learning, the agent fits its parameters using available labeled data before selecting a batch of unlabeled samples to be labeled and integrated into its training set. A well-chosen batch of samples is one that is maximally informative to the learner, such that it can obtain the best hypothesis possible given a fixed labeling budget.

Active learning is well established as an area of machine learning research due to the ubiquity of important real world problems that fit the sample-abundant, label-expensive setting; commonly cited applications range from medical diagnostics [1, 2] to image labeling [3]. Mitigating large sample complexity requirements is particularly relevant for deep neural networks, which have in recent years achieved impressive success on a wide array of tasks but often require considerable amounts of labeled data.

Shifting the focus of active learning to deep neural networks highlights several important problems. For one, most foundational active learning work assumes a convex setting, which is clearly violated by massive nonlinear neural networks. Many of these approaches are computationally expensive, and it is not clear how to adapt them for real-world use [4]. Further, because neural network training is generally expensive, practical active learning algorithms must be able to work in the batch regime, querying $B$ samples at each round of active learning instead of a single point at a time [5].

Despite a long history of active learning research, these constraints draw attention to a need for practical, principled batch active learning algorithms for neural networks. Current state-of-the-art methods, like Batch Active Learning by Diverse Gradient Embeddings (BADGE), perform robustly in experiments, but explanations for its behavior are fairly limited [6]. This drawback makes it unclear how to scale some active learning algorithms into regimes that deviate somewhat from the setting for which they were designed—BADGE, for example, cannot be run on regression problems, and as we show in this paper, performs poorly when used in conjunction with a convex model.

35th Conference on Neural Information Processing Systems (NeurIPS 2021).

This article adopts a *probabilistic perspective* of neural active learning. We view neural networks as specifying a conditional probability distribution $p(y \mid x, \theta)$ over label space $\mathcal{Y}$ given example $\mathcal{X}$, where $\theta$ are the network parameters. This perspective provides theoretical inspiration from the convex regime with which to examine and design neural active learning algorithms. From this viewpoint we motivate and revisit a classic, Fisher-based objective for idealized active selection. We argue that approximately minimizing this objective can be done tractably in the neural regime, despite their overparametrized structure and shifting internal representation. Accordingly, this work helps bridge the divide between algorithms that are performant but not well understood by theory, and those that are theoretically transparent but not computationally tractable.

Experimentally, BAIT offers improved performance over baselines in deep classification problems, a trend that is robust to experimental conditions like batch size, model architecture, and dataset. Crucially, BAIT is general purpose, and can be easily extended to regression settings, where many other algorithms cannot. It further performs well on both regression and classification with convex models, a paradigm in which other algorithms often struggle.

In summary, this paper

- puts neural active learning on firm probabilistic grounding, giving a new, rigorous perspective on the functionality of previously proposed algorithms.

- provides in-depth empirics that elucidate differences between neural and convex regimes, and discusses simplifying assumptions that are sometimes reasonable in the neural case.

- proposes a practical, unifying, high-performing active learning algorithm that leverages these insights in a computationally tractable manner.

## 2   Related work

Active learning is a very well-studied problem [7–9]. There are two main sample selection approaches, diversity and uncertainty sampling, which are successful respectively for large and small batch sizes.

Diversity sampling strategies aim to select batches of data that best represent the space. In a deep learning context, these algorithms typically embed unlabeled samples using the neural network's penultimate layer and select a subset of samples that might act as a proxy for the entire dataset [10, 11]. [12] proposed inducing batch diversity using a generative adversarial network formulation, selecting samples that are maximally indistinguishable from the pool of unlabeled examples.

There is also a rich body of work on batch active learning [13–17]. These methods typically formulate batch selection as an optimization that minimizes an upper-bound on some notion of model loss.

Efficiently adapting parameters to an incrementally larger training set is not an issue in convex settings. Accordingly, with linear models, it is more common to use uncertainty sampling and a batch size of one. A frequently used approach is to query samples that lie closest to the current model's decision boundary, a quantity that's considered inversely proportional to uncertainty [18–20]. Some similar methods offer theoretical guarantees on statistical consistency [9, 21]. Other algorithms quantify uncertainty using the entropy of the predicted distribution over classes, or as the size of the expected gradient induced by observing the label corresponding with a candidate sample [22]. The latter is known to be related to the $T$-optimality criterion in experimental design, but is unable to account for batch diversity.

Similar approaches have been modified for use with neural networks as well. For example, [23] exercise Dropout to sample weights to approximate the posterior distribution over labels, and use it to identify samples that reduce model uncertainty. Adversarial example generation has been used to approximate the distance between a sample and the decision boundary [24]. Model ensembling has also been used to approximate sample uncertainty, where the predictive variance across constituent models can be used to inform a sample selection strategy [25].

There are a variety of algorithms that are meant to combine uncertainty and diversity sampling [26]. This trade-off is sometimes framed as its own optimization problem, for example using a meta-learning approach that hybridizes both strategies [27, 28]. Among these is active learning by learning, which uses a bandit approach to select which query rule to employ at any given round of active learning [28]. BADGE, described in detail in Section 5.1, also combines uncertainty and diversity sampling, and is considered state-of-the-art for deep neural networks.

## 3 Notation and Setup

We consider a standard setup for batch active learning with neural network models, where there is an instance space $\mathcal{X}$, label space $\mathcal{Y}$, and a distribution $D$ over $\mathcal{X} \times \mathcal{Y}$. We use $D_{\mathcal{X}}$ to denote the marginal distribution over the instance space and $D_{\mathcal{Y}|\mathcal{X}}(x)$ to denote the conditional distribution over labels given example $x$. For learning, we are given access to a pool $U = \{x_i\}_{i=1}^n \sim D_{\mathcal{X}}$ of unlabeled examples and we have the ability to request the label for any point $x \in \mathcal{U}$. In the $t^{\text{th}}$ round of batch active learning, we select a collection $\{x_j^{(t)}\}_{j=1}^B \subset U$ of $B$ examples ($B$ is the batch size) and request the labels $y_j^{(t)} \sim D_{\mathcal{Y}|\mathcal{X}}(x_j^{(t)})$ for all examples in the batch. We use these labeled examples to update our neural network model and then we proceed to the next round.

In this setup, the ultimate objective is to achieve low loss on the data distribution $D$, that is we hope our learned parameters $\widehat{\theta}$ nearly minimize $\mathbb{E}_{(x,y)\sim D}\ell(x,y;\widehat{\theta})$, where $\ell$ is some loss function like the cross entropy loss for classification. We always consider this objective in our experiments, but for algorithm development it is helpful to instead consider the *fixed-design* or *transductive* setting, where the goal is to instead minimize $L_U(\theta) = \mathbb{E}_{x\sim U}\mathbb{E}_{y\sim D_{\mathcal{Y}|\mathcal{X}}(x)}\ell(x,y;\widehat{\theta})$, essentially treating the unlabeled samples $U$ as the entire distribution. Note that these two objectives can typically be related by generalization arguments.

## 4 Probabilistic Perspective

We consider neural networks as specifying a probability distribution $p(y \mid x, \theta)$ over the label space $\mathcal{Y}$ given an example $x$, where $\theta$ are the network parameters. Adopting this view, it is most natural to use the loss function $\ell(x,y;\theta) = -\log p(y \mid x, \theta)$, choosing parameters that maximize the likelihood of observed labeled data. In classification problems, for example, we apply the softmax operation to the output of the network and then evaluate the cross-entropy loss with the ground truth label. For regression problems, we use the square loss, which treats the neural network as specifying a Gaussian distribution for each $x$.

**Bayesian linear regression.** As a warm-up, it is illustrative to consider an experimental design setting with Bayesian linear regression. We consider a $d$-dimensional linear regression problem where we assume the parameter vector $\theta^\star$ has prior distribution $\mathcal{N}(0, \lambda^{-1}I)$ and the conditional distribution $D_{\mathcal{Y}|\mathcal{X}}(x) = p(\cdot \mid x, \theta^\star) = \mathcal{N}(\langle\theta^\star, x\rangle, \sigma^2)$ is Gaussian. For any set of labeled data $\{x_j, y_j\}_{j=1}^m$, the resulting maximum a posteriori (MAP) estimate is given by ridge regression with regularizer $\lambda\sigma^2$:

$$\widehat{\theta} = \underset{\theta}{\operatorname{argmin}} \sum_{j=1}^m (\langle x_j, \theta\rangle - y_j)^2 + \lambda\sigma^2\|\theta\|_2^2 \tag{1}$$

In the experimental design setting, we have unlabeled data $U = \{x_i\}_{i=1}^n$ and our goal is to select a set $S \subset U$ of $B$ points so that the resulting MAP estimate has the lowest Bayes risk. Letting $\Sigma = \frac{1}{n}\sum_{i=1}^n x_i x_i^\top$ denote the second moment matrix of the unlabeled data, the Bayes risk is

$$\operatorname{BayesRisk}(S) = \mathbb{E}\left[(\widehat{\theta}_S - \theta^\star)^\top \Sigma(\widehat{\theta}_S - \theta^\star)\right], \tag{2}$$

where $\widehat{\theta}_S$ is the MAP estimate after querying for labels on subset $S$, and the expectation is with respect to the noise in the labels and the prior over $\theta^\star$.

Lemma 1 in the Appendix shows that for a subset $S \subset U$, letting $\Lambda_S = \sum_{x\in S} xx^\top + \lambda\sigma^2 I$, the Bayes risk in this setting is exactly:

$$\operatorname{BayesRisk(S)} = \sigma^2 \operatorname{tr}(\Lambda_S^{-1}\Sigma). \tag{3}$$

Observe that the RHS does not depend on labels, implying that minimizing the RHS over subsets $S$ is feasible and *optimal* selection strategy under this criteria. This also verifies that multiple batches of active learning are not required for Bayesian linear regression, although this observation does not carry forward to the neural setting. BAIT is designed to approximately minimize this objective.

**Classical regime.** An objective similar to (3) also emerges naturally in the analysis of maximum likelihood estimators (MLE) in the convex regime. Here, classical statistical theory posits that the model is *well-specified*, so that there is some parameter $\theta^\star$ such that $D_{\mathcal{Y}|\mathcal{X}}(x) = p(\cdot \mid x, \theta^\star)$ for each $x \in \mathcal{X}$. It is also common to impose regularity conditions including strong convexity of the loss function $L_U(\theta)$ [29, 30]. While these conditions certainly do not hold in the neural setting, BAIT builds on much of this classical technology. The key quantity is the *Fisher information matrix* $I(x; \theta) := \mathbb{E}_{y \sim p(\cdot|x,\theta)} \nabla^2 \ell(x, y; \theta)$ which is known to determine the asymptotic distribution of the maximum likelihood estimator [29]. In many probabilistic models, including linear and logistic regression, the hessian of the loss function does not depend on the label $y$, which we assume going forward.

For active learning in the classical setup, [4] give a two-phase sampling scheme with provably near-optimal performance. In the first phase, the algorithm samples a batch of $B$ points uniformly at random, requests their labels, and optimizes the log-likelihood to obtain an initial estimate $\theta_1$. In the idealized version of the second phase, a batch of $B$ points is chosen to optimize

$$\underset{S \subset U, |S| \leq B}{\operatorname{argmin}} \ \operatorname{tr}\left( \left( \sum_{x \in S} I(x; \theta_1) \right)^{-1} I_U(\theta_1) \right) \tag{4}$$

where $I_U(\theta_1)$ is the Fisher over all samples, $\sum_{x \in U} I(x; \theta_1)$. This combinatorial problem is intractable in general, so [4] instead solve a semidefinite relaxation (SDP). They request labels on the obtained batch $B$ of points and re-fit the model to obtain the final estimate $\widehat{\theta}$. For their setup, they prove the statistical properties of this two-phase estimator are near optimal.

Despite the theoretical properties, solving an SDP is not feasible in high dimensions; instead we provide a new, greedy algorithm for minimizing the objective that is usable in the neural regime. Several other works have also looked at this objective, either from a purely theoretical perspective [4, 31] or via relaxations [32–34]. However, some of these do not ensure batch diversity, and none have been extended to the neural regime.

This formulation essentially generalizes (3), since in linear regression the Fisher information $I(x; \theta)$ is the covariance matrix $xx^\top/\sigma^2$: the objectives in (3) and (4) differ only in their use of the regularizer controlled by $\lambda$. That is, essentially the same objective can be derived from two different perspectives, making it a compelling object for active selection. As such, our starting point for the neural setting is the ideal-but-intractable optimization problem in (4).

## 5  BAIT

Batch Active learning via Information maTrices (BAIT) is inspired by this theory, but adapted to the sequential, neural setting. To do this effectively, several key issues need to be addressed:

1. For neural models, the pointwise information matrix $I(x; \theta)$ is typically extremely large.

2. The internal representation learned by the network changes with each round of active learning, so computation from previous rounds cannot be reused.

3. Solving the objective in Equation (4), as suggested in more theoretical work, is computationally infeasible [4].

Outlined as Algorithm 1, BAIT addresses item 1 in a somewhat standard way, by operating on the last layer of the network [6, 10]. We consider last-layer Fisher matrices $I(x; \theta^L) := \mathbb{E}_{y \sim p(\cdot|x,\widehat{\theta})} \nabla^2 \ell(x, y; \theta^L)$ for last-layer parameters $\theta^L$. Note that in the linear setting $\theta^L = \theta$. Here, if the top-layer representation starts to well-approximate a convex model, then the information geometry induced solely by these parameters can guide active sampling. Further, as we discuss shortly, this top-layer framework gives us a more principled understanding of the empirical success of BADGE.

One more subtle issue (item 2) is the interplay between the changing representation as learning progresses. We address this with an iterative scheme, where the Fisher information matrix is continually recomputed as the algorithm changes its representation during the course of learning.

Rather than solving an SDP, BAIT approximates a solution to Equation (4) using a greedy approach, which we show can be made tractable in both classification and regression settings. At each step of

**Algorithm 1** BAIT

**Require:** Neural network $f(x; \theta)$, unlabeled pool of examples $U$, initial number of examples $B_0$, number of iterations $T$, number of examples in a batch $B$.

1: Initialize $S$ by drawing $B_0$ labeled points from $U$ & fit model on $S$: $\theta_1 = \text{argmin}_\theta \mathbb{E}_S[\ell(x, y; \theta)]$
2: **for** $t = 1, 2, \ldots, T$: {forward greedy optimization} **do**
3:     Compute $I(\theta_t^L) = \frac{1}{|U|} \sum_{x \in U} I(x; \theta_t^L)$
4:     Initialize $M_0 = \lambda I + \frac{1}{|S|} \sum_{x \in S} I(x; \theta_t^L)$
5:     **for** $i = 1, 2, \ldots, 2B$: **do**
6:       $\tilde{x} = \text{argmin}_{x \in U} \text{tr}((M_i + I(x; \theta_t^L))^{-1} I(\theta_t^L))$
7:       $M_{i+1} \leftarrow M_i + I(\tilde{x}; \theta_t^L), S \leftarrow \tilde{x}$
8:     **end for**
9:     **for** $i = 2B, 2B - 1, ..., B$: {backward greedy optimization} **do**
10:       $\tilde{x} = \text{argmin}_{x \in S} \text{tr}((M_i - I(x; \theta_t^L))^{-1} I(\theta_t^L))$
11:       $M_{i-1} \leftarrow M_i - I(\tilde{x}; \theta_t^L), S \leftarrow S \setminus \tilde{x}$
12:     **end for**
13:     Train model on $S$: $\theta_t = \text{argmin}_\theta \mathbb{E}_S[\ell(x, y; \theta)]$.
14: **end for**
15: **return** Final model $\theta_{T+1}$.

the algorithm, the key computation lies in evaluating

$$\tilde{x} = \underset{x \in U}{\text{argmin}} \, \text{tr}((M_i + I(x; \theta_t^L))^{-1} I(\theta_t^L)), \tag{5}$$

where $M_i$ is the Fisher corresponding to samples that have been selected so far.

Unfortunately, the trace function is not submodular, and is thus not well suited for standard greedy optimization. To address this, during each iteration, where the goal is identify $B$ points to query, sampling is done in two stages. For a batch of $B$ points, the first stage greedily oversamples, adding $2B$ samples to the initial batch. In the second stage, BAIT prunes $B$ samples from the batch, better minimizing the objective described in (4). We find that this forward-backward strategy sometimes improves performance over the forward-only alternative (Figure 1). See Algorithm 1 for details. Choosing two as the oversampling factor of two is done for computational reasons, trading-off between computational cost and batch quality. We did not see performance improvements for larger multipliers.

When evaluating the $i$-th sample to include in $S$, the minimization in Equation (5) is efficiently computed using a trace rotation and the Woodbury identity for low-rank inverse updates:

$$\underset{x}{\text{argmin}} \, \text{tr}\left(\left(M_i + V_x V_x^\top\right)^{-1} I(\theta_t^L)\right)$$
$$= \underset{x}{\text{argmin}} \, \text{tr}\left(\left(M_i^{-1} - M_i^{-1} V_x A^{-1} V^\top M_i^{-1}\right) I(\theta_t^L)\right)$$
$$= \underset{x}{\text{argmin}} \, \text{tr}\left(M_i^{-1} I(\theta_t^L)\right) - \text{tr}\left(M_i^{-1} V_x A^{-1} V_x^\top M_i^{-1} I(\theta_t^L)\right)$$
$$= \underset{x}{\text{argmin}} \, \text{tr}\left(M_i^{-1} I(\theta_t)\right) - \text{tr}\left(V_x^\top M_i^{-1} I(\theta_t^L) M_i^{-1} V_x A^{-1}\right)$$
$$= \underset{x}{\text{argmax}} \, \text{tr}\left(V_x^\top M_i^{-1} I(\theta_t^L) M_i^{-1} V_x A^{-1}\right),$$

Figure 1: A comparison between forward and forward-backward greedy approaches for BAIT. Here we show a simple active learning experiment using an MLP and MNIST data [35], and samples are acquired in batches of size 10 for 50 rounds. See Section 6 for more details.

where $A = I + V_x^\top M_i^{-1} V_x$ is an easily invertible $k \times k$ matrix. Here $V_x$ is a $dk \times k$ matrix of gradients, where each column is scaled by the square root of the corresponding prediction: $V_x V_x^\top = I(x, \theta^L)$. This formulation keeps us from having to compute and store all candidate $I(x; \theta^L)$, drastically decreasing the algorithm's memory footprint. The trace rotation step, placing $V_x$ as the leading term instead of $M_i^{-1}$ is essential, as it avoids computing a new $kd \times kd$ matrix for each $x$. As a practical matter, on all datasets we consider in Section 6, this allows us to compute the trace contribution of all candidate samples simultaneously on a modern GPU.

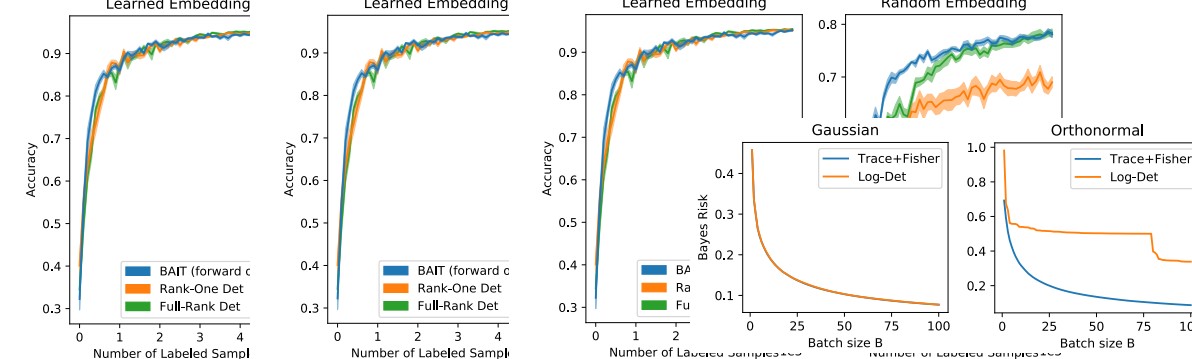

Figure 2: Linear classification on different representations of MNIST data. **Left:** Learned features, similar those from a neural network. **Right:** A random, uninformed projection, simulating the raw features a convex model may have to use.

Figure 3: Bayesian linear regression simulations comparing BAIT and determinantal maximization. In both cases the data have poorly conditioned covariance matrices with quadratic spectral decay. Determinantal maximization exploits this in the Gaussian case, but not the orthonormal case. BAIT performs well in both settings.

After the minimizer $x$ is found, updating $M_i^{-1}$ is done simply via the same Woodbury identity, $M_{i+1}^{-1} = M_i^{-1} - M_i^{-1} V_x A^{-1} V_x^\top M_i^{-1}$, and the algorithm proceeds to identify the next sample.

**Regression.** In the regression setting we are able to further reduce the amount of required computation. Let $x^L$ denote the penultimate layer representation induced by $f(x; \theta)$. For linear models $x^L = x$. In Appendix Section A.3, we show that a $k$-output regression model trained to minimize squared error has pointwise Fisher $I(x; \theta^L) = (x^L)(x^L)^\top \otimes \widehat{\Sigma}^{-1}$, where $\widehat{\Sigma}$ is the noise covariance of the estimator. Using this fact, the regression version of the Fisher objective is

$$\mathrm{tr}\left( \left( \sum_{x \in S} I(x; \theta^L) \right)^{-1} I_U(\theta^L) \right) = k \, \mathrm{tr}\left( \left( \sum_{x \in S} x^L (x^L)^\top \right)^{-1} \left( \sum_{x \in U} x^L (x^L)^\top \right) \right). \tag{6}$$

The full derivation can be found in Appendix A.3.1. This observation greatly simplifies the minimization in Equation (5), allowing us to use only rank one matrices $x^L (x^L)^\top$ in place of the rank $k$ matrices in the classification setting. The procedure is written explicitly in Appendix A.4.

### 5.1 BADGE comparison

By comparison, BADGE, a recently proposed, state-of-the-art active learning classification algorithm, aims to select a batch of samples that are likely to induce large and diverse changes to the model [6]. This is done by representing each candidate sample $x \in U$ as $g_x = \nabla \ell(x, y = \widehat{y}; \theta^L)$, the $d$-dimensional last-layer gradient that would be obtained if the most likely label according to the model, $\widehat{y}$, were observed. BADGE selects a batch of samples that have large Gram determinant in this space.

The intuition behind BADGE is that sampling proportionally to the Gram determinant of these hallucinated gradients trades-off between uncertainty and diversity; a batch of gradient embeddings that produce a large Gram determinant will need to be both high magnitude (corresponding to model uncertainty) and linearly independent (corresponding to batch diversity). It is worth noting that while BADGE sampling is *motivated* by determinantal point process (DPP) sampling, the actual BADGE algorithm only uses a rough approximation to this procedure.

Still, from the perspective of BAIT, the "gradient embedding" used in BADGE is a single column of the $dk \times k$ matrix $V_x$, but not scaled by $\sqrt{p_i}$. These embeddings can correspondingly be thought of as rank-one approximations for $I(x; \theta)$. BAIT trades BADGE's determinantal sampling for a trace minimization ($A$-optimality). This substitution is essential because the determinantal approach is unable to accommodate for $I(\theta)$, as $\mathrm{argmax}_x \det(I(x; \theta)^{-1} I(\theta)) = \mathrm{argmax}_x \det(I(x; \theta)^{-1})$ for any $I(\theta)$.

Thus, BAIT offers two main advantages over BADGE. First, it considers the entire rank-$k$ pointwise Fisher, catching potentially useful information that's ignored by BADGE. Second, BAIT incorporates the Fisher over all samples $I(\theta)$, a term we show to be essential for minimizing risk and bounding MLE error. Crucially, because BADGE identifies this vector as corresponding to the most likely label according to the model, and this only makes sense in classification settings, it is unable to handle regression problems, a regime to which BAIT naturally extends.

**Comparing objectives.** These observations make it clear that BAIT is more general than BADGE, but it is not obvious which of the aforementioned algorithmic extensions is most important for boosting performance. Figure 2 directly compares those objectives for batch sample selection. Specifically, we run three variations: greedily maximizing the determinant of the rank-one BADGE

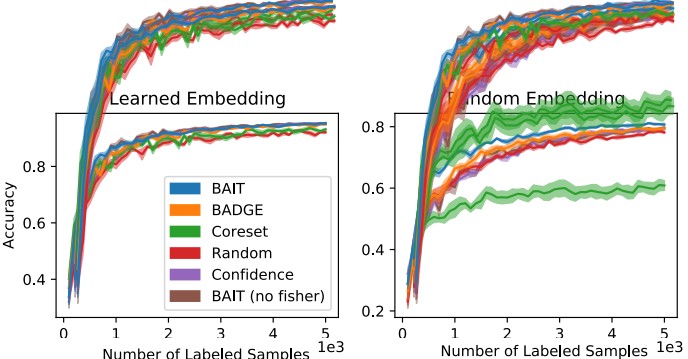

Figure 4: The same plots as Figure 2, but comparing BAIT to baseline active learning algorithms. We include BAIT without $I(\theta^L)$ for a clearer comparison. BAIT most drastically outperforms baselines on the uninformed representation, where high-norm directions are not necessarily most discriminative.

gradient embeddings, greedily maximizing the determinant of the full-rank Fisher, and the BAIT approach, taking into consideration both the full-rank pointwise Fisher matrices and $I(\theta^L)$. The two determinantal algorithms are written formally in the Appendix as Algorithm 2 and Algorithm 3, and can be made efficient by taking advantage of Woodbury identities. Here BAIT uses only forward greedy optimization, rather than both forward and backward, to ensure a fair comparison.

We study two simple projections of the MNIST dataset. In one, we fit a two-layer MLP on 50% of the training data, and embed the remaining 50% using the first layer. We perform active learning in this 128-dimensional space on the unseen 50% of examples, selecting 50 batches of size 10 in sequence.

We then conduct a similar experiment, but instead of using a learned representation, we use a random (Gaussian with mean zero and unit variance) matrix to project samples into 128 dimensions, a setting in which, unlike in the learned representation, the largest directions are not necessarily the most discriminative. Note that this representation allows us to control feature dimensionality but mimics the typical convex learning paradigm, where features are fixed, not conditioned on labels, and not controllable by the learner.

In both plots, the BAIT objective outperforms determinantal objectives. This effect is more drastic for the uninformed embedding, where the plot suggests that the full-rank pointwise Fisher is more useful than its low-rank counterpart for late-stage performance, and that $I(\theta^L)$, as used in BAIT, is especially beneficial for early stage performance.

**Synthetic experiment.** We conduct a small synthetic experiment to demonstrate the value of incorporating the Fisher information matrix into the acquisition strategy. In Figure 3 we plot the exact Bayes Risk (3) in the Bayesian linear regression setup described in Section 4 as a function of the batch size $B$ for both BAIT and the greedy determinant maximization strategy. Here we consider two distributions in $d = 100$ dimensions. In the left plot data are generated from a Gaussian distribution with diagonal covariance with quadratic spectral decay $\Sigma_{ii} \propto 1/i^2$. On the right, the distribution is supported only on the standard basis, with probabilities that decay quadratically $p_i := \mathbb{P}[x = e_i] \propto 1/i^2$. Note that both distributions have identical and poorly conditioned covariance $\Sigma$ (recall (3)).

This allows us to highlight the value of the Fisher matrix and how it leads to robust performance across data distributions. Indeed, we see that in the Gaussian case, both the BAIT strategy (called "Trace+Fisher" in the figure) and the determinental maximization strategy ("Log-det") perform almost identically. However, BAIT significantly outperforms the alternative in the orthonormal case. This occurs because the latter does not exploit the occurrence probabilities $p_i$ and in fact simply selects the coordinates in a cyclic fashion. On the other hand, the optimal strategy focuses effort on the high-probability coordinates, which is exactly captured in the Fisher matrix.[1]

## 6 Experiments

In this section we detail extensive experiments that highlight the generality and performance of BAIT. We consider three settings: linear classification, deep classification, and regression. Throughout these sections, we compare BAIT to several recently proposed and classic active learning approaches.

Among these, we consider BADGE, CORESET, CONFIDENCE, and RANDOM sampling. BADGE, as mentioned earlier, is a state-of-the-art approach that trades off between diversity and uncertainty by approximately sampling a batch of points that have high Gram determinant when represented as a

---

[1]Note that in the orthonormal case, both greedy optimization algorithms are in fact optimal for their respective combinatorial problems.

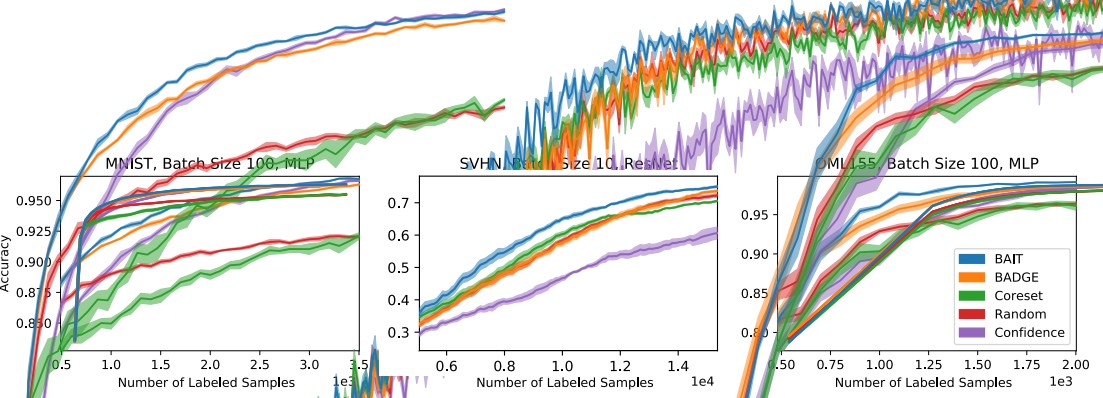

Figure 6: Three deep active learning experiments with different model architectures, datasets, and batch sizes. **Left:** An MNIST experiment, using a batch size of 100 and an MLP. **Center:** Active learning on the SVHN dataset with an 18-layer ResNet and a batch size of 10, smoothed for clarity (unsmoothed plot in the Appendix). **Right:** Active learning on the OpenML dataset 155 using an MLP and a batch size of 100. Here we zoom in on disriminative regions of learning curves.

gradient. CORESET represents items using the model's penultimate layer representation, then samples a batch that describes the space well. CONFIDENCE sampling selects the $n$ points for which the model is least confident, measured by $\max f(x; \theta)$. RANDOM draws $n$ points uniformly at random.

## 6.1 Linear Classification

Like BADGE and CORESET, BAIT caters to efficiency in part by only considering the last layer of the network to select a new batch. Despite this linear assumption, both CORESET and BAIT are unable to perform well outside of the neural regime.

This subsection revisits the simplified setting described in Section 5.1 and Figure 2, involving both informed and uniformed representations of MNIST. In the learned representation, performance differences between algorithms is relatively subdued, with BAIT, BADGE, and CONFIDENCE among the highest-performing agents. However, in the unstructured, random representation, the are stark differences in accuracy. While controlling for dimensionality, this representation mimics the convex case, where the model is not able to control how data are represented. Here, BAIT outperforms baseline approaches by a large margin (Figure 4).

Among these comparisons, we include a simplified version of BAIT, which omits the Fisher term $I(\theta)$, resulting in an objective that has been explored by [36]. This approach performs on par with other baselines, suggesting that it is the inclusion of $I(\theta)$ that allows BAIT to succeed even in difficult, poorly structured feature spaces. This experiment further highlights a potential cause of the success of these baselines, as the penultimate-layer representation will behave more like what's described here as a learned representation than a random representation. Still, the following subsection shows BAIT outperforming baselines in deep classification.

## 6.2 Deep Classification

We now turn to our main experiments, active learning for classification with neural networks. This subsection provides extensive results for the above algorithms across a wide array of settings.

We consider three datasets. Using an MLP, we perform active learning on both MNIST data and OpenML dataset 155. We also use the SVHN dataset [37] of color digit images with both an MLP and an 18-layer ResNet. Last we explore the CIFAR-10 object dataset [38] with a ResNet. All dataset-architecture pairs are experimented with at three batch sizes—10, 100, and 1000. MLPs include a single hidden ReLU layer of 128 dimensions.

All ResNets are trained with a learning rate of 0.01, and all other models (including linear models shown earlier) are trained with a learning rate of 0.0001. We fit parameters using the Adam variant of SGD, and use standard data augmentation for all CIFAR-10 experiments. Like other deep active learning work, we avoid warm-starting and

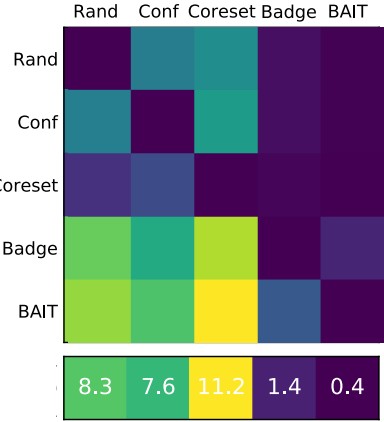

Figure 5: A pairwise comparison plot. Element $i\,j$ roughly corresponds to the number of times algorithm $i$ outperforms algorithm $j$ by a statistically significant degree. Columwise averages are given at the bottom, where a lower number corresponds to a higher-performing algorithm.

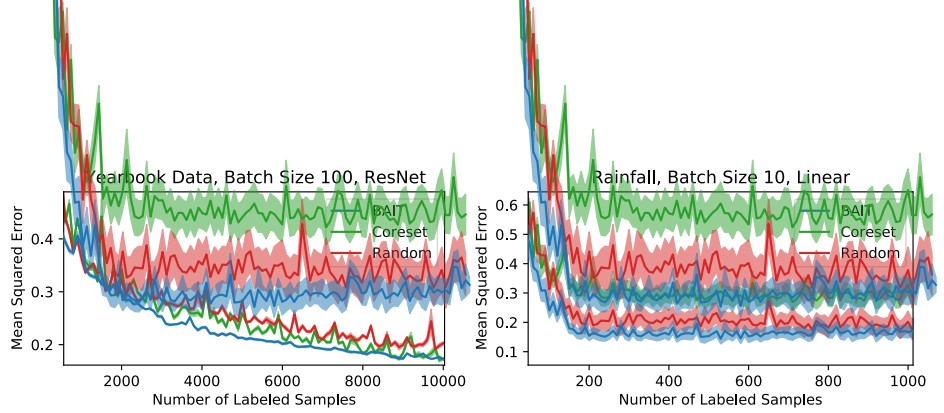

Figure 7: Two regression experiments with varying architectures. **Left:** Active regression using an 18-layer ResNet, predicting the year in which American yearbook photos were taken. **Right:** A linear model used to predict rainfall from meteorological features.

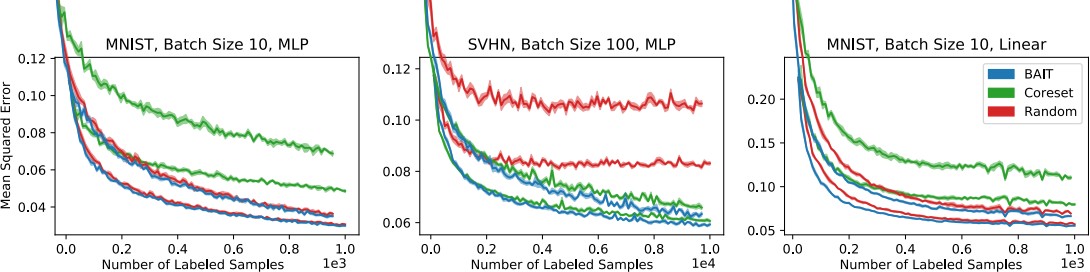

Figure 8: Three regression experiments with varying architectures. **Left:** Active regression using an MLP, MNIST data, and a batch size of 10. **Center:** Active regression using SVHN data and a batch size of 100. **Right:** the same as the leftmost plot, but using a linear model instead of an MLP.

retrain model parameters from a random initialization after each query round [5]. Each learner is initialized with 100 randomly sampled labeled points, and each experiment is repeated five times with different random seeds. Shadowed regions in plots denote standard error. More empirical details can be found in Appendix Section C.

Figure 6 zooms in on the discriminative regions of learning curves corresponding to three different settings. While the relative performance of baseline algorithms varies somewhat across scenarios, BAIT is consistently as good or better than the highest-performing approach. Full learning curves are presented in Appendix Section C.1.

Due to the volume of settings investigated, we present aggregate results using the analysis approach of [6]. For each experiment, we note the round $r$ of active learning for which random selection first obtains accuracy within 1% of its final accuracy. We then checkpoint each algorithm at exponential intervals up to $r$, that is, we log each labeling budget $L$ for which $L_k = M_0 + 2^k B \leq r$, for batch size $B$ and number of seed samples $M_0$. At each $L$ in a given experiment, we compute the $t$-score, $t = \frac{\sqrt{N}\hat{\mu}}{\hat{\sigma}}$, where $N$ is the number of samples, between each pair of algorithms $i \neq j$ as

$$\hat{\mu} = \frac{1}{N} \sum_{l=1}^{N} (e_i^l - e_j^l), \qquad \hat{\sigma} = \sqrt{\frac{1}{N-1} \sum_{l=1}^{N} (e_i^l - e_j^l - \hat{\mu})^2},$$

where $e_i^l$ and $e_j^l$ denote the $l$-th accuracy respectively corresponding to algorithms $i$ and $j$ at labeling budget $L_k$. We then perform a two-sided $t$ test, where algorithm $i$ is said to outperform algorithm $j$ if $t > 2.776$, and vice versa if $t < -2.776$, marking a significant difference ($p < 0.05$).

This formulation allows us to construct a *pairwise penalty* matrix over all conducted experiments. The matrix has as many rows and columns as there are considered algorithms (five); if algorithm $i$ outperforms algorithm $j$ for some experiment at some labeling budget, the corresponding element $i$ $j$ of the matrix is incremented by $1/z$, where $z$ is the total number of labeling budgets considered for that experiment.

The resulting plot is given in Figure 5, which aggregates results over all conducted experiments, and which suggests BAIT significantly outperforms baseline approaches. We also include columnwise averages, which give a holistic perspective on algorithm performance.

We show more pairwise plots of this type in Appendix Section C.2, breaking up results by batch size and and architecture type. These figures all suggest BAIT is higher-performing than baseline approaches across environments.

### 6.3 $L_2$ Regression

Although deep learning is most commonly discussed within a classification framework, recent work has successfully applied deep learning in regression settings as well, with important scientific applications including areas like physical, biological, and chemical modeling [39, 40]. It is therefore important to develop active learning algorithms that are flexible enough to be applied in these domains.

Figure 7 presents active learning results using two different model architectures, two different batch sizes, and two different datasets. In the first, we train an 18-layer ResNet to predict the year in which photos from an American yearbook were taken [41]. To do this successfully, the

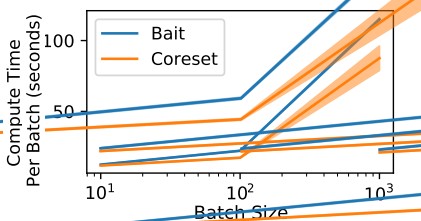

Figure 9: In the regression setting, the rank-one reduction of greedy selection in BAIT makes the approach only slightly slower than Coreset.

model must learn to correlate trends in photography and fashion with a time period. Here labels were Z-scored, so error is not measured in terms of year. In the second, we use meteorological features and a linear model to predict the amount of rainfall in Austin, Texas [42]. Note that there are few active learning algorithms made with regression in mind—the CONFIDENCE and BADGE algorithms are omitted here, as they rely on a notion of uncertainty that requires a classification environment.

In Figure 8, we show a few regression active learning experiments that have been synthetically adapted from the classification setting. We treat SVHN and MNIST data as having $k$ continuous outputs, regressing onto one-hot encodings of their labels.

Similar to the classification case, the relative performance of baseline approaches shuffles between environments. Batch active learning in regression is challenging, with simple random sampling often being surprisingly effective. Still, regardless of which baseline is highest performing, BAIT consistently performs as well or better on both linear and non-linear regression tasks. Further, because BAIT can be reduced to rank-one calculations (Equation 6), it is relatively efficient, and takes about as long to run as CORESET (Figure 9).

## 7 Discussion

This article studies neural active learning from the theoretical perspective of maximum likelihood estimators, a viewpoint that sheds new light on the performance of previous approaches. We proposed BAIT, a generalized, high-performing, and tractable approach to neural active learning that makes use of this perspective, showing that a more classical approach is tractable and effective with modern neural architectures. We demonstrated that BAIT is successful in both convex and non-convex scenarios, and for both classification and regression settings.

It is worth noting that, while tractable, the classification version of BAIT is more computationally intensive than BADGE—roughly $k$ times slower to select a sample to include in a batch (in seconds, though total run times are largely dominated by retraining models after each batch acquisition [5]). This added computation is well justified in active scenarios for which the cost of label acquisition is high relative to the cost of computation. To trade-off between computation requirements and performance, one could estimate the Fisher using only the lowest-norm $r < k$ columns of $V_x$, catching the more descriptive components of the Fisher. We leave the analysis of such an approach as an avenue for future work.

## 8 Acknowledgements

Sham Kakade acknowledges funding from the National Science Foundation under award #CCF-1703574.

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
