# OpenReview forum: "Gone Fishing: Neural Active Learning with Fisher Embeddings"
_NeurIPS.cc/2021/Conference — NeurIPS 2021 Poster_

### Official Review · Reviewer_7jff · 2021-06-29

**Rating:** 6
**Confidence:** 4

**Summary:**

The paper tackles active learning with deep learning models, focusing primarily on the problem that labels need to be requested in batches due to the large training cost. The authors propose to rely on the Fisher information and introduce a greedy approach to choose each set of points to be labelled efficiently.


**Limitations And Societal Impact:**

The main limitation discussed is the increased cost per sample selection. A societal impact is not discussed (chosen as not applicable in the checklist).

The societal impact follows the general trend of deep learning approaches here. Better active learning strategies allow for more efficient labeling, giving us less wasted human time as a direct societal benefit. To a second degree, it then depends on whether the labeled data is used for beneficial or harmful tasks.


**Main Review:**

## Originality
While the specific approach of the Fisher matrix is new and the paper mentions that it is revisiting a "classic, Fisher-based objective" (l42), it is lacking a proper discussion of the prior work of using such approaches, which have been explored before (e.g. back to MacKay, (1992) who relied on it for active learning with neural networks).

Similarly, the citations and general presentation require a lot of improvements. The main competitor, BADGE, is mentioned first in the introduction and repeatedly throughout the paper. Yet, it is never connected with a reference leaving the reader to guess its source (ref 35 is mentioned, but only independently). The same applies to a second baseline, coreset, which is never connected to reference 8. Similarly, at other locations, e.g. while one could, if need be, argue that MNIST/SVHN/Cifar10/ResNet/Adam have become common knowledge, it would still be cleaner to cite them. However, the same argument does not apply to something like "OpenML dataset 155" (l293), which I would consider to be nonstandard and unknown by the average reader.


## Quality
The method is presented in a somewhat biased way. E.g. with repeated claims in the regression setting, both that other active learning methods are lacking in that direction, and even deep learning usually being only discussed for classification (l335); a consistent improvement, but small in absolute numbers, is sold as "large margin" (l282) and "drastic" (caption of Figure 4).
Apart from such outliers and the citation problems mentioned above, the presentation is well done. The method is explored on several data sets (synthetic and real), several architectures and batch size settings, with extensive further code being provided.

The comparisons are primarily against deterministic methods (Badge, Coresets), lacking a comparison against the growing literature of Bayesian Deep Learning. E.g. the Gal et al. (2017), Kirsch et al. (2019) (the latter of which actively targets batch active learning). What do the authors expect performance-wise compared to such approaches that actively include model uncertainty as well as predictive uncertainty, together with query rules adapted for Bayesian Neural Nets (e.g. (Batch)BALD)?


## Clarity
Apart from the details mentioned in the two sections above, the paper is written clearly and well organized.

## Significance
The significance of the proposed approach is difficult to judge. In relative performance, the proposed model shows consistent improvements upon the baselines (especially the extended experiments in the appendix). Yet, in absolute numbers, the improvements upon simple approaches such as randomly picking new points or a vague confidence metric remain small. This leaves the question open as to how much of a practical influence it will have. It would be interesting to see the performance of the proposed approach to a highly redundant setting where most of the unlabeled points are irrelevant, punishing random query strategies and requiring a clever query, such as the proposed, to select the few highly relevant points.



## Further Questions/Minor comments/typos
- l32 claims a lack of principled batch active learning algorithms. Yet, the paper cites both Sener et al. and Kirsch et al., which try to tackle exactly this task in a principled way.
- Can the authors comment on the repeated claim of lack of approaches for active learning in the regression setting (e.g. l49-50)? A lot of Bayesian uncertainty-based methods are directly applicable to the regression setup. Similarly, Active Learning's closely related sister field, Bayesian Optimization, regularly tackles continuous and discrete parameter spaces.
- l66 claims references [11-15] to be about batch AL but separate from deep learning. Yet [15] explicitly tackles the task of Bayesian Deep Learning.
- Error: References [29], [30] are the same paper
- $V_x$ appears out of thin air in the equation starting in l184 without a proper introduction (and in a typo misses one subscript after the first equality)
- Error: The right plot in Figure 5 mentions OML155 in its title, while the caption claims it to be Cifar-10.
- Section 6.3 claims deep learning to be most commonly discussed within a classification framework, but it is, of course, also applied to a lot of regression tasks. E.g. staying in the image world, the whole area of depth estimation etc.
- Figure 1 shows a consistent improvement of the proposed forward/backward approach vs the strict forward, yet the text claims that this holds only "sometimes" (l180). Are there setups where it always performs worse or is it only due to random initializations that sometimes cause it to perform worse in some runs?



___
(MacKay, 1992: Information-based objective functions for active data selection)


**Time Spent Reviewing:**

7

---

> ### Author Response · Authors · 2021-08-10
> **response to reviewer 7jff**
>
> Thank you for your detailed review and feedback.
>
> Originality:
>
> 1. We agree that more classical, Fisher-based active learning algorithms should be mentioned. We should and will include a thorough discussion of the MacKay paper, as it is foundational work for active learning with neural networks.
>
> 2. We further agree that baselines should be cited earlier in the paper, and that presentation would be improved if datasets, architectures, and optimizers are cited upon their first mention. We will rectify this in the camera-ready version.
>
> Quality:
>
> 1. As you say, “large margin” and “most drastic” is perhaps language too strong to describe a performance boost that never exceeds ten percentage points. We will weaken our claims in the next revision.
>
> 2. BADGE is actually not a deterministic algorithm. It relies on k-means++ for selection, which is a stochastic, farthest-first traversal algorithm. Also, the approach is built explicitly to include, and trade-off between, uncertainty and diversity. This point is mentioned in the BADGE paper’s appendix.
>
> BatchBALD was excluded only because of its computational expense. Using the popular code available from https://github.com/BlackHC/BatchBALD, selecting the first batch of 10 SVHN images using a P100 GPU and six CPU cores, takes about fifteen minutes. Selecting a batch of 100 takes more than four and a half hours.  These numbers don’t include the added time induced by having to train models with dropout.
>
> Significance:
>
> 1. Thanks for this nice question about experimenting with redundancy! Actually this is very close to what we do in the synthetic experiment in the right panel of Figure 3. Here the data distribution is supported on the standard basis vectors $e_1,...,e_d$ but the probabilities are highly skewed as $Pr[x = e_j] \propto j^{-2}$. As such, one should prioritize sampling the first few directions (as our objective does), but neither random sampling nor the determinant maximization do this. In fact, determinant maximization is essentially random sampling without replacement here and you can see that its performance (measured via the “BayesRisk”) is quite poor.
>
>
>
>
>
>
>
>
>
>
>
>
>
>
>
>
>
>
>
>
>
>
>
>
>
>
>
>
>
>
>
>
>
>
>
>
>
>
> Further Questions:
> 1. As you mention, some work we cite in deep active learning propose solutions that are not unprincipled, and even offer mathematical justification. We will adjust our phrasing to better respect this.
>
> 2. It’s true that Bayesian active learning approaches like BatchBALD can in principle handle regression problems, but their compute requirements, as mentioned above, precludes them from being compared with.
>
>     Along these lines, Bayesian Optimization becomes difficult when the data dimensionality exceeds (roughly) 10 dimensions. Accordingly, Bayesian Optimization is used heavily for problems like hyperparameter tuning, but not used for active learning. ([Shahiri et. al], "Taking the human out of the loop: A review of Bayesian optimization.") says this degradation in empirical performance with increasing dimensions is likely due to over-exploration.
>
> 3. Our usage of the word "sometimes" with respect to the improvement of the forward-backward approach over the forward-only approach was only meant to convey that we had not run all active learning experiments with both forward only and forward-backward selection.

---

> > ### Comment · Reviewer_7jff · 2021-08-18
> > **Thank you for your answers**
> >
> > Thank you for your answers to my questions!
> >
> > If the clarifications are included in the final version as promised my score remains stable and I still recommend acceptance.

---

### Official Review · Reviewer_GaQR · 2021-07-15

**Rating:** 6
**Confidence:** 3

**Summary:**

The paper introduces an algorithm (BAIT) for batch active learning, designed for and tested on neural networks. It centres around an objective in eq 4, argmin_select trace(I_select^-1, I_all), with I as the Fisher information matrix of the final layer parameters of a neural network, an objective that has been previously derived/analysed in simpler settings. The contribution here is applying it to deep neural networks. It draws links with a current SOTA method, BADGE, and shows slight performance benefits over it.

**Limitations And Societal Impact:**

There’s a short paragraph at the end of the paper outlining the additional computation complexity of the method. Considering the addition of an extra page in this year’s conference, I think this should be expanded.

**Main Review:**

I’m borderline on this paper. I do find the method interesting, and the empirical performance seems to be in line with the current SOTA method BADGE. However, I believe the presentation of the method needs improving, and there are various claims that I am uncomfortable with, including interpretation of results. Overall, I think the paper could be very strong following a sober revision.

__Method Structure/Clarity__

I found the motivating theory (section 4) of the method confusing. It begins by considering the case of Bayesian linear regression to derive eq 3, but then jumps to pulling out eq 4 from Chaudhuri paper, which it says the paper will optimise. What is the use of beginning with eq 3? The setting considered in the paper is neither Bayesian, nor linear, nor generally regression. I think it would be more worthwhile to discuss eq 4 and the Chaudhuri paper in fuller detail. E.g. it’s said this was shown to be ‘near optimal’ under their set up. What was this set up? What assumptions were required? How do these translate to the neural set up?

Several times (including abstract) it’s stated that we’re optimising a bound on the MLE error. But nowhere in the technical sections did I see any bound. Under what conditions is it a valid bound (I'm assuming just the linear setting)?

As a smaller point, the paper gives the impression that viewing the output of a network as an estimated probability distribution is a novel perspective – I’d argue that this is the default position.

In general, I didn’t come away feeling like I have much intuition for why it’s a good idea to choose a set to minimise eq 4.


__Claims and Wording__

Several times it’s claimed that the method addresses problems of non-convexity in neural systems (including the abstract). But actually, the Fisher information matrix is only ever considered for the last layer parameters, with the majority of the network used as a feature extractor. This is ok as a methodology, but it feels like misleading advertising.

The paper claims to achieve SOTA results. This is a bit of a hard sell when one looks over the full learning curves in the appendix, where curves for BAIT vs BADGE for CIFAR10 and SVHN are pretty well indistinguishable. The zoomed regions in figure 5 seem rather cherry-picked. I’m unsure if very slight advantages justify BAIT’s increased computational demands (k times). As I reviewer, I don’t demand SOTA results if the method is interesting enough in itself, which I think is the case here, and would prefer the authors to have a more considered results discussion.

There does seem to be a performance improvement in the linear setting (‘random embeddings’), but this is of less interest given the paper’s title and main objective. In some ways it seems unfair to give these figures (2, 4) such prominent position.


__Experiments__

Being able to apply the method in regression settings is a great advantage, but the experiments tested simply took the classification datasets (MNIST etc) already used and converted them to regression. This feels like an opportunity missed, as there there are plenty of interesting regression tasks, e.g. depth regression, counting problems, rainfall prediction...

To test the linear setting, data is 'randomly projecting data into d dimensions’ (rather than using features extracted by a neural net). I think the use of ‘random’ is a bit misleading here as it seems to have used some dimensionality reduction technique? I didn’t spot how this projection was done, but it seems important to describe it.

The algorithm works by selecting 2B examples, then discarding B of these. I’d be interested to see ablations/rationale for how this number was selected. Could the algorithm be improved with a larger number?


__Minor__
- Seems like section headings have been crushed up, eg line 262
- Algorithm 1, perhaps move ‘forward greedy optimization’ to line 5?
- Subscripts and superscripts missing on eqs on line 184
- Think redundant /2 in line 498


**Time Spent Reviewing:**

5-6

---

> ### Author Response · Authors · 2021-08-10
> **response to reviewer GaQR**
>
> Thank you for your feedback.
>
> Method Clarity:
>
> 1. To be clear, we want to emphasize that the objective derived from the Bayesian perspective is almost identical to that from the MLE perspective, or rather a specific instantiation of the MLE perspective (note that in linear regression, the inverse covariance matrix is the inverse Fisher information). Thus, because essentially the same objective can be derived from two perspectives, we believe it to be a compelling quantity for active selection. We will clarify the similarity between these two perspectives in the final version.
>
> 2. The setup of the Chaudhuri paper is convex, so it does not straightforwardly extend to the neural regime. Rather than claiming that the theoretical results from Chaudhuri directly apply, we argue that the resulting object is worth investigating in the non-convex setting.
>
> 3. You are correct in assuming that the BAIT objective bounds MLE error in the linear setting. To improve clarity, we will restate this theoretical result from the Chaudhuri paper in our appendix
>
> 4. To build intuition for the BAIT objective, it is helpful to work from the simplest case, where $I(\theta)$ is the identity matrix. In this case, the BAIT objective is very similar to determinantal maximization, where we are primarily encouraged to select a batch of points that are very spread out (in the sense that their second moment matrix has large eigenvalues). In the general case, where $I(\theta)$ is not the identity, it provides a warping/transformation that is adapted to the data distribution, so that we prioritize the “important” directions of the data, in addition to encouraging sufficient spread. You can see this in action in our synthetic experiments in Figure 3.
>
> Claims and Wording:
>
> 1. You are correct, BAIT does not address the non-convexity in neural models. We agree that the wording chosen suggests it does, apologize for the confusion, and will correct this mistake in the updated version.
>
> 2. As you mention, BAIT sometimes only does as well as the best baseline, rather than outperforming it, and the zoomed-in figures of the main body are meant to highlight the situations in which BAIT does outperform other approaches. This is not unusual for active learning, where margins of improvement are often small (see the appendix of the BADGE paper for an example). The SOTA claim is meant to be in the average case, much like the BADGE paper. We will clarify this in the camera ready.
>
> Experiments:
>
> 1. We agree that the paper would be improved by including more regression results. Please see the comment addressed to all reviewers about the two new environments added, including the rainfall prediction task suggested in your review.
>
>
> 2. The random projection is done by multiplying the data by a $q \times d$ matrix, where $q$ is the intrinsic dimensionality of the data. The entries of this matrix are gaussian random numbers with mean zero and unit variance. This is indeed a valid dimensionality reduction technique, but it does not account for labels in doing the projection. The goal of this experiment is to compare “informed” (chosen specifically for the supervised task at hand) and “uninformed” (chosen without any particular task in mind) representations while controlling for dataset dimensionality. We will improve our explanation of this in the final version.
>
> 3. Increasing the forward selection beyond $2B$ would indeed obtain a lower objective value, but it would also increase computational expense. $2B$ was chosen to strike a balance between performance and computational cost. As you mention, the additional page allocated this year could be used for a more thorough discussion about computational complexity, and we will do so in the context of this choice.

---

> > ### Comment · Reviewer_GaQR · 2021-08-19
> > **update**
> >
> > Thanks for your response, and I appreciate the new regression experiments. Conditional on tempering the wording around some of your claims, I'm updating my score slightly.

---

### Official Review · Reviewer_gqfA · 2021-07-16

**Rating:** 7
**Confidence:** 4

**Summary:**

The paper introduces a new principled active learning algorithm called BAIT, which uses the Fisher information matrix to select informative samples. The approach uses a greedy approximation to solve the global batch acquisition problem. Compared to other approaches, it also works for regression.

The paper provides ablation and gives a new perspective on BADGE, another active learning algorithm.

Multiple experiments both for classification and regression are provided.

**Limitations And Societal Impact:**

There is no mention of societal impact, which given what other papers have written is not a big loss, as this method does not have any specific new societal impact compared to existing AL methods.

The paper does not mention limitations per-se beyond the high computational load. I wonder how dependent this method is on the # of latent dimensions.

**Main Review:**

### Originality

The approach seems interesting. It focuses on using Fisher information matrices to select points that will minimize the overall loss on the pool set: $\underset{S \subset U,|S| \leq B}{\operatorname{argmin}} \operatorname{tr}\left(\left(\sum_{x \in S} I\left(x ; \theta_{1}\right)\right)^{-1} I_{U}\left(\theta_{1}\right)\right)$.

The algorithm to select points is novel in particular. The objective is not submodular, so greedy selection does not have guarantees. Instead, the paper greedily selects more samples than required and then prunes the batch again. (I particularly like the forward (expansion) and backward (pruning) approach. Could this be repeated multiple times to improve the candidate batch?) However, there is no theoretical analysis available of this approach yet.

Fisher Information has been applied to Active Learning with DNNs previously, e.g. ["Active Deep Learning with Fisher Information for Patch-wise Semantic Segmentation", Sourati et al, 2018 (https://pubmed.ncbi.nlm.nih.gov/30450490/)](https://pubmed.ncbi.nlm.nih.gov/30450490/).

The main contribution lies in the novel algorithm (BAIT) that does not require semi-definite programming to find batches and in examining BAIT empirically in deep learning.

Moreover, the paper provides a new perspective on BADGE, an earlier active learning algorithm. It shows that BADGE makes stronger approximations than BAIT. An ablation shows that BAIT outperforms BADGE. I like that BAIT does take into account the whole pool set (through the overall information matrix).

### Quality

Overall, the submission seems sound. The experiments in the main paper are convincing and necessary ablations are provided. The method is very well motivated and deduced.

While Figure 6 in the paper convincingly shows that BAIT outperforms all other methods, the respective plots in the appendix are less clear than the ones in the main paper. In particular, it seems that the proposed method, BAIT, does not really outperform BADGE on SVHN and CIFAR-10 by much at all. It would be good to swap the experiments on MNIST with another higher-dimensional or more challenging dataset. Maybe CINIC-10 might provide more conclusive results (https://paperswithcode.com/dataset/cinic-10).

Results are also presented for OpenML dataset 155. However, I do not know this dataset and it is not introduced. (That said, citations for MNIST, SVHN and CIFAR-10 are also missing.)

Lastly, the regression experiments use a rather contrived setup. Fitting one-hot encodings is a very specific objective. However, I cannot suggest anything better.

### Clarity

The paper is very well written. The introduction and related work section are great. As noticed, a few more citations ought to be added. The introduction is also rather light on citations.

### Significance

The results appear very significant assuming they perform better than previous methods, which I am not fully convinced of yet.
The proposed algorithm appears significant by itself as the idea of selecting and pruning alternatively is novel. However, given the shown results on CIFAR-10 and SVHN, it would not be clear whether this method is truly SOTA, and whether to recommend this method over BADGE as a baseline in the future.

The application to regression is significant and definitely an improvement over other methods (like BADGE) which only work with classification. However, the regression experiments are not as convincing.

### Summary

The paper introduces a new AL method, which is well motivated and clearly communicated. This method is not limited to classification and also works for AL for regression.

The plots in the appendix are not as clear as I'd like, and I would like to verify that BAIT outperforms BADGE manually as well instead of trusting Figure 6.

### Questions

1. Could you provide experimental results on another higher-dimensional dataset? (E.g. CINIC-10)
2. Could you smooth the plots (using a Gaussian window or similar) to make the SVHN and CIFAR-10 results more easily interpretable?
3. Could you provide a less contrived regression experiment?
4. How dependent is this method on the # of latent dimensions?

I would be happy to increase my score if any or all of the above questions are answered.

### Typos

1. [29] and [30] in the bibliography refer to the same paper.
2. (4): it is not clear what $I_U$ exactly is at first.
3. l.184: what is $V_x$? It has not been introduced.
4. Sec 5.1: Figure 2 is never directly referenced in the text.
5. Figure 5's caption says CIFAR-10 while the plot says OML155.
6. l. 316: what is N? Number of trials?

Disclaimer: I did not check the proofs in the appendix in great detail.


**Time Spent Reviewing:**

7

---

> ### Author Response · Authors · 2021-08-10
> **response to reviewer gqfA**
>
> Thank you for your review.
>
> Comments:
>
> 1. Indeed, the forward-backward approach could be done iteratively to better minimize the BAIT objective, although this of course comes with a higher computational cost. We chose to do this only once for solely computational reasons.
>
> 2. Thank you for pointing out the paper by Sourati et. al. This work appears to consider only the Fisher for the batch, $I(x, \theta)^{-1}$, and excludes $I(\theta)$, the Fisher over all data. The brown line in Figure 4 corresponds to this objective, where we show the importance of including $I(\theta)$. We will better explain this objective in the context of Sourati et. al. in the camera-ready copy.
>
> 3. The absolute magnitude of performance difference between active learning algorithms is unfortunately often quite small. This can be seen in the appendix of the BADGE paper as well, where they somewhat remedy the problem by zooming in on discriminative regions for each of the plots. We will do this as well in the camera-ready copy.
>
> 4. We agree that citing datasets would improve the clarity of the work, and will do so in the revised copy.
>
> Significance / Summary:
>
> 1. As you mention, there are many dataset / architecture / batch size combinations for which BAIT is only marginally better than, or about as performant as, the best baseline. Determining whether it’s worth spending the extra computation on BAIT, rather than just doing BADGE is difficult, but BAIT does appear to be SOTA in aggregate (Figure 6). We will zoom in on appendix figures, similar to the BADGE paper appendix, to improve clarity.
>
> Questions:
>
>
> 1. As stated above, we have added two new regression experiments, one on image data and another on meteorological data. The former is high dimensional, and both are less contrived than what’s included in the original submission. On your recommendation, we also worked the CINIC-10 dataset into our repository. The rebuttal period is not enough time to obtain full results for these data, but we will continue to work with it.
>
> 2. Smoothed plots do seem to show clearer disparity between algorithms. As an example, we’ve included a smoothed version of the second pane of Figure 7, viewable at https://imgur.com/a/p5xLJJe, and would be happy to include these in the final version (either as figure replacements or as supplementary appendix figures). In this linked figure, the first pane is the original plot and the second pane is the smoothed version.
>
>
> 3. Note that there is some variance in latent dimensionality in the presented experiments. In the ResNet experiments, for example, the latent dimensionality is 128. For OpenML 155, when using a linear model, the latent dimensionality is the same as the data dimensionality, 10. OML-155 was chosen because it was also used in the BADGE paper.
>
>     The relative performance of active learning algorithms seems to be robust to this parameter, but it should be noted that, since BAIT does need to maintain a matrix that grows with the number of latent dimensions, it will not be feasible to store this if d is extremely large. We will make this more clear in the camera-ready.

---

> > ### Comment · Reviewer_gqfA · 2021-08-18
> > **Re**
> >
> > Dear authors,
> >
> > thank you very much for running additional experiments and answering all my questions.
> >
> > I am particularly pleased that the smoothed plots are an improvement for readers. It will strengthen the paper to include them.
> >
> > The results on the two additional regressions look great too. It would be nice to have these experiment setups/datasets introduced into the literature. It has been mentioned to me before that there were no good high-dim datasets with low-dim regression outputs for AL, so I am very happy to see that you have found two.
> >
> > As my questions have been answered satisfactorily, I will raise my score. I'm looking forward to the improved paper.
> >
> > Best wishes

---

### Author Response · Authors · 2021-08-10
**response to all**

Thank you all for your time and feedback.

Following the suggestion of several reviers, we have carried out two new regression experiments. In one, the goal is to predict the amount of rainfall on a given day from a set of meteorological features (https://www.kaggle.com/grubenm/austin-weather). Here we compare methods using a linear model and a batch size of 10. Results are viewable at https://imgur.com/a/5EV1w2q.

In another, we use a dataset of American high-school yearbook images, where the task is to predict the year in which each photo was taken (https://arxiv.org/abs/1511.02575). To do this task well, the model must pick up on trends in fashion and photography, and correlate them with a time period. Here we use an 18-layer ResNet and a batch size of 100. Results are available at https://imgur.com/a/F4ZykfU. Note that we z-scored the labels of these data, so the MSE is not measured in terms of years.

Mistakes noticed by multiple reviewers:


1. We will better introduce $I_u$ and $V_x$ in our revision. For a given sample $x$, $V_x$ is the $dk \times k$ matrix of gradients, where each column is a gradient corresponding to a possible label, and where each column is scaled by the square root of the corresponding prediction. $V_xV_x^\top = I(x, \theta)$.


2. The rightmost plot in Figure 5 is OML-155, not CIFAR-10. Sorry for the confusion!


3. We will correct all typos and citation errors mentioned. Thank you for bringing these to our attention.

---

### Decision · Program_Chairs · 2021-09-27

**Decision:**

Accept (Poster)

**Comment:**

Reviewers agreed that this is a good contribution to NeurIPS, *provided* that the additional experiments and clarifications provided by the authors during the discussion are included to the paper.